# MicroRNA-210-3p is transcriptionally upregulated by hypoxia induction and thus promoting EMT and chemoresistance in glioma cells

**Hong Liu**[1], **Changjin Chen**[1], **Jinhao Zeng**[1], **Ziyi Zhao**[1☯*], **Qiongying Hu**[2☯*]

**1** Hospital of Chengdu University of Traditional Chinese Medicine, Chengdu, China, **2** Department of Laboratory Medicine, Hospital of Chengdu University of Traditional Chinese Medicine, Chengdu, China

☯ These authors contributed equally to this work.
* zhaoziyi925@163.com (ZZ); qiongyinghu@163.com (QH)

**Data Availability Statement:** All relevant data are within the manuscript and its Supporting Information files.

## Abstract

### Background

Glioma is the most common and lethal form of brain cancer. It is highly malignant and is often characterized by chemoresistance and radioresistance, which are thought to mainly result from hypoxic microenvironments. Various tumour-promoting and tumour-suppressing microRNAs (miRNAs) have been identified in gliomas; however, it is still largely unknown how miRNAs are modified by hypoxia and subsequently affect glioma. In this study, we examined the expression of miR-210-3p, a well-characterized miRNA that responds to hypoxia in glioma cell lines.

### Methods

The expressions of miR-9 and miR-210-3p were analysed by using qPCR. Cell viability was measured by performing CCK-8 after eechinomycin treatment or introduction of miR-210 for 24 or 48 h. The correlation of HIF-1α expression with TGF-β were analysed using the REMBRANDT database. The biomarkers of EMT, including E-cadherin, N-cadherin and Vimentin, were detected by western blot. Apoptotic cell death was measured by performing Annexin V-FITC/PI double staining followed by flow cytometry.

### Results

We found that miR-210-3p was induced by a mechanism dependent on the hypoxia-induced transcriptional activity of HIF-1α. Then we established a positive association between the HIF-1α and TGF-β expression levels, and miR-210-3p upregulation induced TGF-β expression, indicating that hypoxia-induced HIF-1α activity upregulated TGF-β via miR-210-3p upregulation. Hypoxia-induced miR-210-3p activity was found to promote EMT by upregulating TGF-β, which subsequently enhanced the invasive ability in U87-MG cells. We further confirmed that miR-210-3p induced chemoresistance to TMZ in U87-MG cells via TGF-β upregulation under hypoxic conditions.

**Funding:** Ziyi Zhao. 82074298 for Ziyi Zhao The General Program (Key Program, Major Research Plan) of National Natural Science Foundation of China http://isisn.nsfc.gov.cn/egrantweb/ The funders had no role in study design, data collection and analysis, decision to publish, or preparation of the manuscript.

**Competing interests:** The authors have declared that no competing interests exist.

## Conclusion

These results help to reveal the potential regulatory mechanisms of hypoxia-induced miR-210-3p expression that affect malignant behaviors and chemoresistance via TGF-β upregulation in glioma cells.

## Introduction

Glioma is the most common and lethal brain tumor; it is highly malignant and is often characterized by chemoresistance and radioresistance [1]. Glioma is characterized by high mortality and recurrence rates [2], lethal invasiveness [3], and strong angiogenesis in hypoxic microenvironments [4]. A common therapeutic strategy consists of surgical removal, radiotherapy and chemotherapy [5]; however, glioma patients often have poor prognoses mainly due to malignant biological behaviors [6], indicating that glioma cells can migrate to and invade sites away from the primary tumor mass. Epithelial-to-mesenchymal transition (EMT), which is characterized by loss of the epithelial marker (E-cadherin) along with overexpression of mesenchymal markers (N-cadherin and Vimentin) [7], is a well-researched physiological process that results in more robust invasive or metastatic phenotypes [8]; furthermore, EMT is thought to limit total surgical resection and to contribute to therapeutic resistance, which can lead to tumor recurrence in several cancers [9, 10], including in glioma [11]. Many studies have focused on the mechanism of EMT-induced metastasis in glioma; however, the effects of EMT on glioma remain largely unknown.

Hypoxia is well-accepted as a promoting factor in cancer progression, and it contributes to malignant behaviors and metastasis [12]. Poor vasculature leads to a limitation in the delivery of oxygen and nutrients, frequently inducing necrosis in the interior regions of solid tumors. Hypoxia has also been reported to regulate miRNAs, small, single-stranded non-coding RNAs that have multiple critical roles in physiological processes in tumor cells under hypoxic conditions [13]. In gliomas, microRNAs have been identified and characterized as critical regulators in the malignant progression [14]. As the most responsive and influential miRNA, miR-210-3p is positively regulated by hypoxia-induced factor 1α (HIF-1α) [15–17], and it post-transcriptionally regulates several genes. In this manner, miR-210-3p upregulation mediated by HIF-1α exerts critical effects on cell cycle control, apoptosis, and aberrant regulation of cell morphology, polarization and malignant behaviors [18, 19]. However, the potential effects of miR-210-3p and its responsiveness to hypoxia remain largely unknown in glioma.

HIF-1α, which has been postulated to be a hallmark of hypoxia treatment, tightly regulates transforming growth factor-β (TGF-β) and its downstream signaling, resulting in multiple biological effects, including tumorigenesis, progression and metastasis [20–22]. TGF-β has been shown to be a master regulator of the initiation and maintenance of EMT phenotypes in many kinds of cancers [23], and persistent hypoxia exposure induces acquisition of the EMT phenotype via multiple mechanisms, one of which is transcriptional activation of TGF-β via HIF-1α. It is also believed that miRNAs that respond to hypoxic conditions may be involved in these interactions and exert biological effects.

Based on these observations, we hypothesized that hypoxia-responding miRNAs may be involved in regulating a network of physiological processes in glioma cells. We demonstrate here that hypoxia-induced miR-210-3p induction in glioma cells promotes EMT via TGF-β upregulation and induces chemoresistance, which indicates its tumor promoting roles in glioma.

## Material and methods

### Cell culture and treatment

Human glioma cell lines, U87-MG (from unknown origin), A172 and HS683 were obtained from American Type Culture Collection (ATCC, Manassas, VA, USA) and stored in liquid nitrogen in our laboratory. For each cell line, they were stored in liquid nitrogen before 3–5 passages. Cells were maintained in Dulbecco's modified Eagle's medium high glucose (DMEM-H) containing 10% Fetal bovine serum (FBS, Life Technologies, Grand Island, NY, USA), 100 U/mL penicillin and 100 μg/mL streptomycin, and kept in a 5% $CO_2$ incubator at 37˚C. Medium were replaced and cells were passaged every two or three days.

To achieve hypoxia, cells were maintained in a modular incubator chamber (Billups Rothenberg Inc., Del Mar, CA) containing 5% $CO_2$, 95% $N_2$ for 24 to 48 h. To achieve HIF-1α stabilization, 100 μM $CoCl_2$ for 24 to 48 h to mimic hypoxia.

To block transcriptional activity of HIF-1α, cells were pretreated with 1 ng/ml Echinomycin (EC), a transcriptional activity inhibitor of HIF-1α, for 4 h before following treatment.

To mimic TGF-β induced EMT, cells were treated with 5 ng/ml TGF-β for 24 to 48 h before following treatment.

### CCK-8 assay

To evaluate the cell viability, a CCK-8 Cell Counting Kit (CCK8; Dojindo, Kunmamoto, Japan) was employed according to the manufacturer' instruction. Briefly, cells were plated at a density of $2\times10^4$ cells per well in 96-well plates and allowed to attach overnight. After desired treatment, 10 μL of CCK-8 detecting reagent was added into each well for another 4-hour incubation at 37˚C. Then, absorbance at 450 nm was recorded in an EnSpire® Multimode Plate Readers (PerkinElmer, China). The experiments were done in pentaplicate and repeated three times.

### Reverse-transcriptional quantitative PCR (RT-qPCR)

Total RNA was extracted using TRIZol (Life Technologies, Grand Island, NY, USA) according to the manufacturer's instructions.

To detect TGF-β and β-actin mRNA levels, the reverse-transcriptional Kit (Life Technologies, Grand Island, NY, USA) was employed followed by the manufacturer's instruction. Then the cDNA was used as template under the following conditions: 35 cycles of 95˚C for 10 seconds (s); 60˚C for 1 minute (min) on an ABI7500 (Applied Biosystems, Foster City, CA, USA). The primers used for qPCR were as follows: β-actin forward, `5'-CATGTACGTTGCTATCCAGGC-3'` and reverse, `5'-CTCCTTAATGTCACGCACGAT-3'`; TGF-β forward, `CTGACGGCCACGAACTTCC`; and reverse, `5'- GCACTGACATTTGTCCCTTGA-3`. To detect miR-210-3p, miR-9 and U6, All-in-OneTM miRNA qRT-PCR Detection Kit (GeneCopoeia, Guangzhou, China) was used followed by manufacturer's instruction. The approved primers for miR-210-3p (Cat. No.: HmiRQP0317), miR-9 (Cat. No.: HmiRQP0825) and HsnRNA U6 (Cat. No.: HmiRQP9001) were all bought from GeneCopoeia. The qPCR results were analyzed and expressed relative to the CT (threshold cycle) values and then converted to fold changes; 2.0-fold change was considered significant.

### Western blot

Mouse anti-human HIF-1α antibody (Cat.: ab1), rabbit anti-human E-cadherin antibody (Cat.: ab1548), mouse anti-human N-cadherin antibody (Cat.: ab98952), mouse anti-human Vimentin antibody (Cat.: ab8069), rabbit anti-human TGF-β1 antibody (ab92486) and rabbit

anti-human β-actin antibody (Cat.: ab179467) are primary antibodies were bought from Abcam (Cambridge, England) and diluted at 1:2000.

Total protein was prepared using RIPA buffer (Thermo Scientific, Waltham, MA, USA) followed the manufacturer's instruction. Protein concentration was measured by performing BCA staining (Sigma–Aldrich, St. Louis, MO, USA) and 20 μg total protein of each sample was fractionated using 8–16% SDS-PAGE gel, and transferred to PVDF (Millipore, Billerica, MA, U.S.A) membranes. After transferring, blotted membranes were blocked with 5% milk/TBS buffer for 30 min on a shaker, and this was followed by an incubation with primary antibodies at 4˚C overnight. Followed by three washes with PBS-T (containing 0.1% Tween-20), HRP-conjugated secondary antibodies were incubated with PVDF membranes for 1 hour at room temperature. Goat anti-mouse IgG H&L (HRP) (Cat.: ab97040) and goat anti-rabbit IgG H&L (HRP) (Cat.: ab7090) secondary antibodies were employed for specific primary antibody. Blots were developed using Pierce$^{TM}$ ECL Western Blotting Substrate (Thermo Scientific, Waltham, MA, USA) according to the manufacturer's instructions.

## Transswell assay

To evaluate the invasive ability, transwell migration assay was employed using a 24-well transwell chemotaxis chamber technique (Millipore, Billerica, MA, USA). Briefly, DMEM (500 μL) with 10% FBS was added in the lower chamber. $1 \times 10^4$ cells suspended in 200 μL medium were placed into the upper chamber (pore size, 8 μm) coated with 100 μl Matrigel (Millipore, Billerica, MA, USA). The plate was then incubated for 24 h at 37˚C in a humidified atmosphere with 5% $CO_2$. The Matrigel was removed and its upper surface was wiped away with a cotton swab to remove the unmigrated cells and fixed with 4% paraformaldehyde for 15 min followed by staining with 0.1% crystal violet for 10 min. After 3 washes with ice-cold PBS, the number of cells per view was counted in randomly under a light microscope (BL-AC10DS, Olympus, Tokyo, Japan). Each assay was performed in triplicate wells.

## Transfection of miR-210-3p mimics, inhibitor or siRNA target to TGF-β mRNA

SiRNA was employed for knocking down the expression of TGF-β. Introduction of miR-210-3p mimics or inhibitor was performed to regulate miR-210-3p expression.

SiTGF-β (ON-TARGETplus Human TGFBI siRNA SMART pool; L-019370-00-0005, J-019370-06-0002 and J-019370-08-0002) and control nonspecific siRNA (ON-TARGETplus Non-targeting Control Pool; D-001810-10-05) were purchased from Dharmacon (Lafayette, CO, USA).

MiR-210-3p mimics and miR-210-3p inhibitor were obtained from Ambion (Austin, TX, USA).

The transfection was performed using Lipofectamine 2000 (Life Technologies, Grand Island, NY, USA) according to the suggested concentration of manufacturer (Ribobio Co. Ltd, Guangzhou, China).

## Annexin V-FITC/PI double staining

To evaluate apoptotic cell death, the annexin V-FITC/PI double staining was performed according to the manual of Annexin V-FITC/PI apoptosis detection kit (Life Technologies, Grand Island, NY, US). For each sample, $1 \times 10^6$ cells were collected and stained with 5 μl Annexin V-FITC and 10 μl PI staining solution for 30 min in the dark at room temperature, and then the binding buffer was added to 500 μl of total volume and analyzed by performing flow cytometry on 3 laser Navios flow cytometers (Beckman Coulter, Brea, CA, USA). For each sample, $1 \times 10^4$ events were acquired.

## Statistical analysis

All data are expressed as the mean±SD. One-way ANOVA followed by Bonferroni's multiple comparison test were used for comparisons among experiments groups. The correlations were determined by a Pearson's coefficient of correlation. All data analysis was performed using GraphPad Prism 5 software (GraphPad Inc., La Jolla, CA). A $p$-value$<0.05$ was considered statistically significant.

## Results

### MiR-210-3p is induced under hypoxic conditions via HIF-1α transcriptional activity

To evaluate the effects of hypoxia on the miR-210-3p expression level in glioma cell lines, including U87-MG, A172 and HS683, cells exposed to hypoxia for 24 and 48 h were used for quantitative PCR, and miR-9, which does not respond to hypoxic conditions, was used as a negative control [24, 25]. As shown in Fig 1A, hypoxia exposure significantly upregulated the miR-210-3p expression levels at 24 and 48 h in all three glioma cell lines; however, it did not affect miR-9 expression. To confirm whether the transcriptional activity of HIF-1α is involved in hypoxia-induced miR-210 upregulation, we cotreated cells with 1 ng/ml EC, and the miR-210 expression was measured 48 h after hypoxia exposure. As shown in Fig 1B, cotreatment with echinomycin abolished miR-210 upregulation after hypoxia exposure or $CoCl_2$ treatment, demonstrating that HIF-1α transcriptional activity is critical for miR-210 upregulation. By considering that hypoxia exposure affected miR-210 similarly, we focused on the effect of hypoxia on U87 proliferation. Because HIF-1α plays critical roles under ischemia/hypoxia conditions [26], we further analyzed the effects of hypoxia and $CoCl_2$ and the subsequent stimulated HIF-1α transcriptional activity on cell viability. Both hypoxia and $CoCl_2$ exposure significantly decreased cell viability after 24 and 48 h. Addition of echinomycin significantly increased cell viability after 48-hour treatment (Fig 1C).

To further confirm the effect of HIF-1α induced by hypoxia on proliferation, we performed high content screening (HSC)-based cell viability assay. As it is shown in Fig 2A, consistently, Hypoxia exposure decreased cell viability, which was reversed by inhibition of HIF-1α transcriptional activity (Fig 2A). Cell cycle distribution was further analyzed by PI staining followed by flow cytometric assay, and expectedly, HIF-1α critically blocked entry of cell cycle (Fig 2B).

### Hypoxia and CoCl2-induced HIF-1α upregulated TGF-β

By considering that HIF-1α and TGF-β is tightly associated, and HIF-1α is tightly regulated by TGF-β [27, 28], we searched for correlation between HIF-1α and TGF-β in Repository for Molecular Brain Neoplasia Data (REMBRANDT, E-GEOD-68848, http://www.ebi.ac.uk/arrayexpress/) database, and found that TGF-β protein was mildly positive associated with HIF-1α expression (Fig 3A). To confirm the effects of hypoxia on TGF-β, glioma cell lines were exposed to hypoxia or $CoCl_2$ for 48 h and employed for TGF-β mRNA analysis. As it is shown in Fig 3B, hypoxia-induced upregulation of TGF-β was reversed by echinomycin cotreatment in all these three glioma cell lines. Expectedly, addition of 5 ng/ml TGF-β obviously induced HIF-1α, which indicates the positive feed-back loop between HIF-1α and TGF-β, that is induced by hypoxia or $CoCl_2$ (Fig 3C and 3D, left panel). In normoxia condition, introduction of miR-210-3p mimics significantly upregulated TGF-β mRNA level, without being disturbed by echinomycin supplement (Fig 3D, right panel), demonstrated the positive association between miR-210-3p and TGF-β. We also introduced miR-210-3p into normoxia-

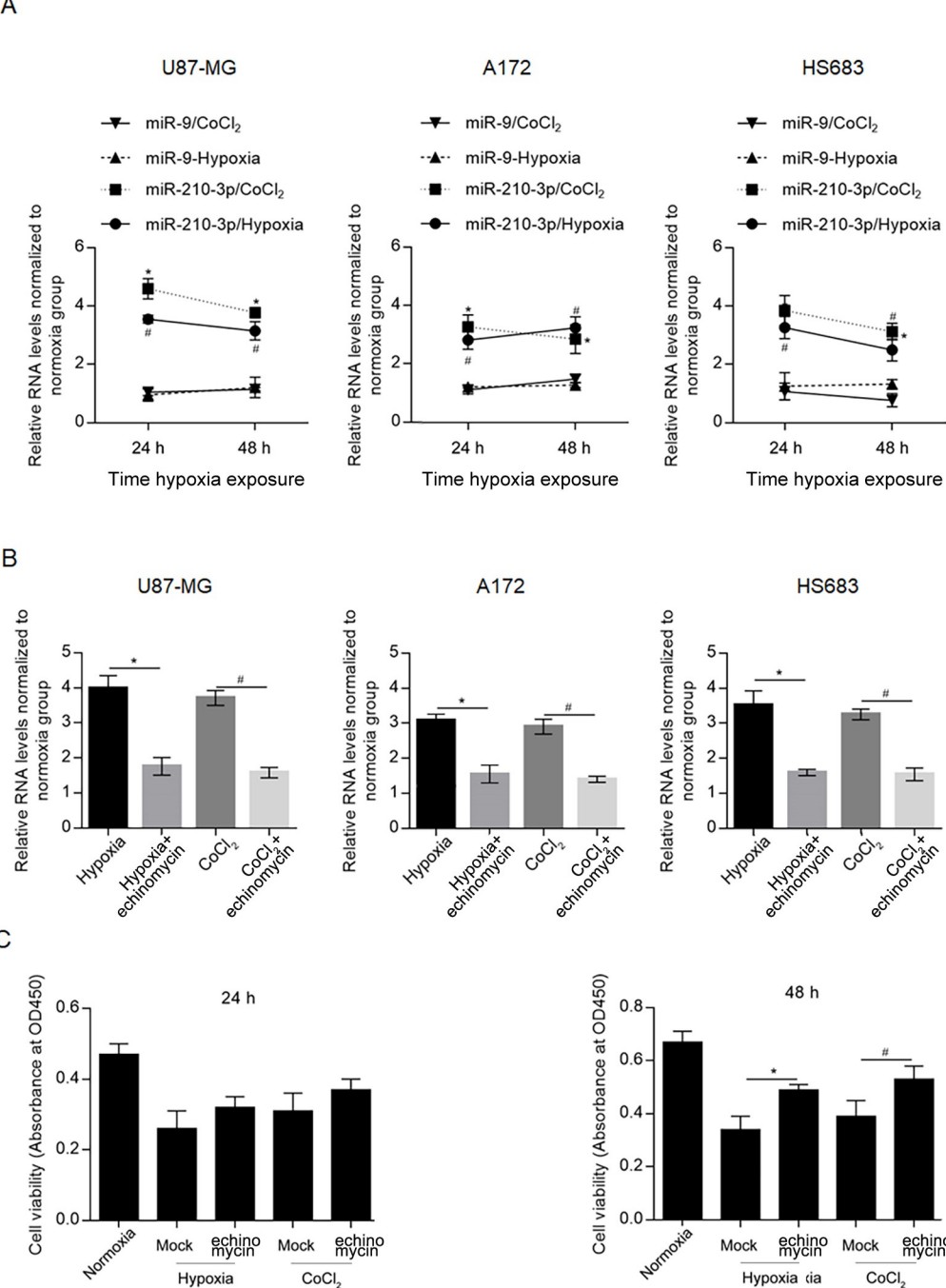

**Fig 1. Hypoxia exposure upregulated miR-210-3p via transcriptional activity of HIF-1α.** A. After hypoxia or $CoCl_2$ exposure for 24 and 48 h, the expressing levels of miR-9 and miR-210-3p were evaluated by RT-qPCR. $^*P<0.05$, vs. 24h normoxia group; $^{\#}P<0.05$, vs. 48h normoxia group. Square represents miR-210-3p/$CoCl_2$ group; Triangle represents miR-9/hypoxia group; reverse triangle represents miR-9/$CoCl_2$ group; circle represents miR-210-3p/hypoxia group. B. After co-treatment with 1 ng/ml echinomycin, the expressing levels of miR-9 and miR-210-3p in glioma cells were detected. $^*P<0.05$, vs. hypoxia group; $^{\#}P<0.05$, vs. $CoCl_2$ group. C. After echinomycin treatment or introduction of miR-210 for 24 or 48 h, cell viability was measured by performing CCK-8. $^*P<0.05$, vs. hypoxia+echinomycin group; $^{\#}P<0.05$, vs. $CoCl_2$+echinomycin group.

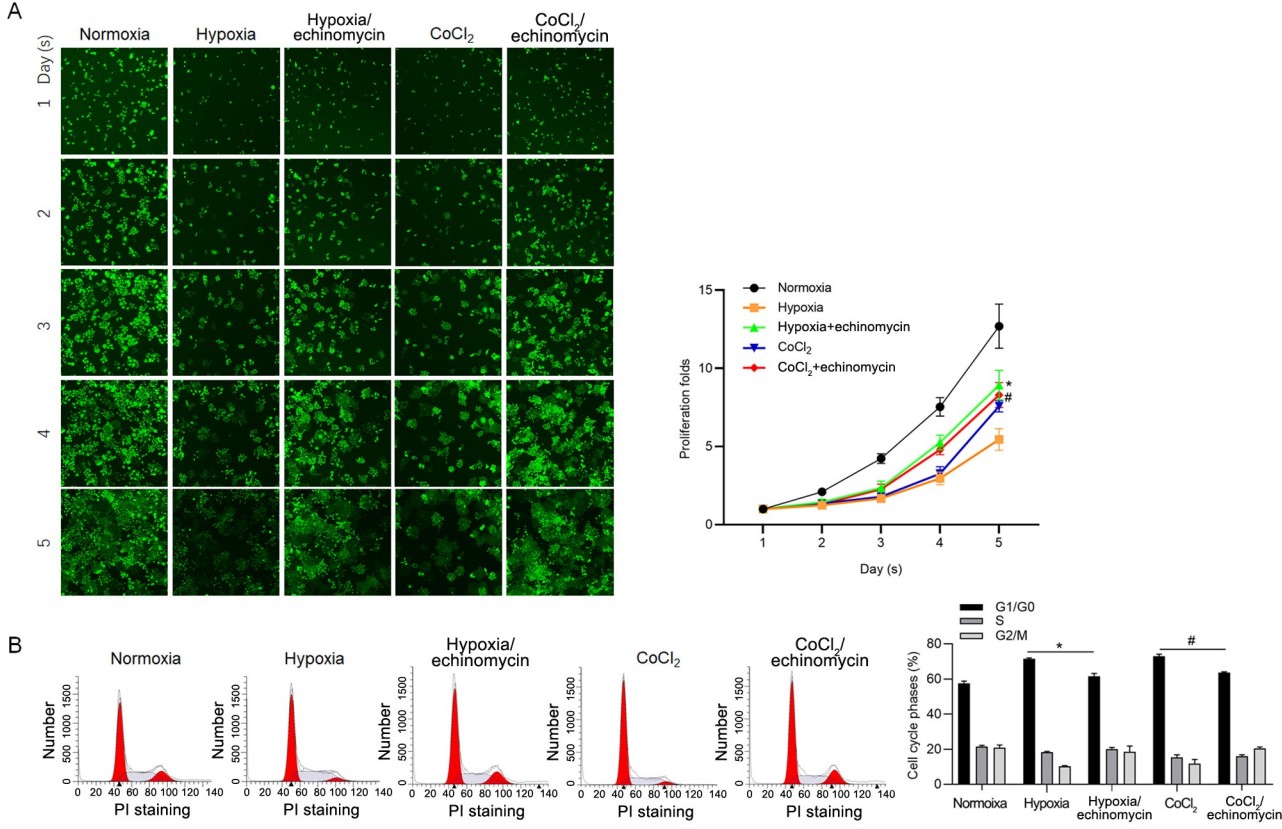

**Fig 2. The effect of transcriptional activity of HIF-1α on proliferation.** A. Cell viability was measured by performing high content screening (HSC)-based assay. B. Cell cycle distribution was analyzed by PI staining followed by flow cytometric assay. *P<0.05, vs. hypoxia group; #P<0.05, vs. CoCl₂ group.

or hypoxia-exposed cells, it was shown that (Fig 3E), the effect of hypoxia on TGF-β mRNA is similar with that of introduction of miR-210-3p mimics, indicated that hypoxia induced upregulation of TGF-β mRNA is mainly via upregulating miR-210-3p.

### Introduced miR-210 mimics upregulated TGF-β and promoted malignant behaviors independent on transcriptional activity of HIF-1α

To confirm the regulation of miR-210 on TGF-β1 mRNA, miR-210 mimics was transfected, while miR-9 mimics was also transfected as a negative control. Efficient introduction of miR-210 or miR-9 was detected which was not affected by addition of echinomycin (Fig 4A). Expectedly, introduction of miR-210 mimics, but not miR-9 mimics, significantly upregulated TGF-β1 mRNA level, which was not affected by inhibition of HIF-1α transcriptional activity (Fig 4B). Then we focused on the effects of miR-210 mimics on malignant behaviors, including cell cycle distribution, invasion and colony formation. As shown in Fig 4C, miR-210 mimics decreased proportion of G1/G0 phase independent on the presence of HIF-1α transcriptional activity. The promoting effects of miR-210 mimics on invasion and colony formation were also observed (Fig 4D and 4E). Taken together, it is demonstrated that the effects of miR-210 on transcriptional level of TGF-β1 and malignant behaviors of glioma cells, is independent on the HIF-1α transcriptional activity induced by hypoxia.

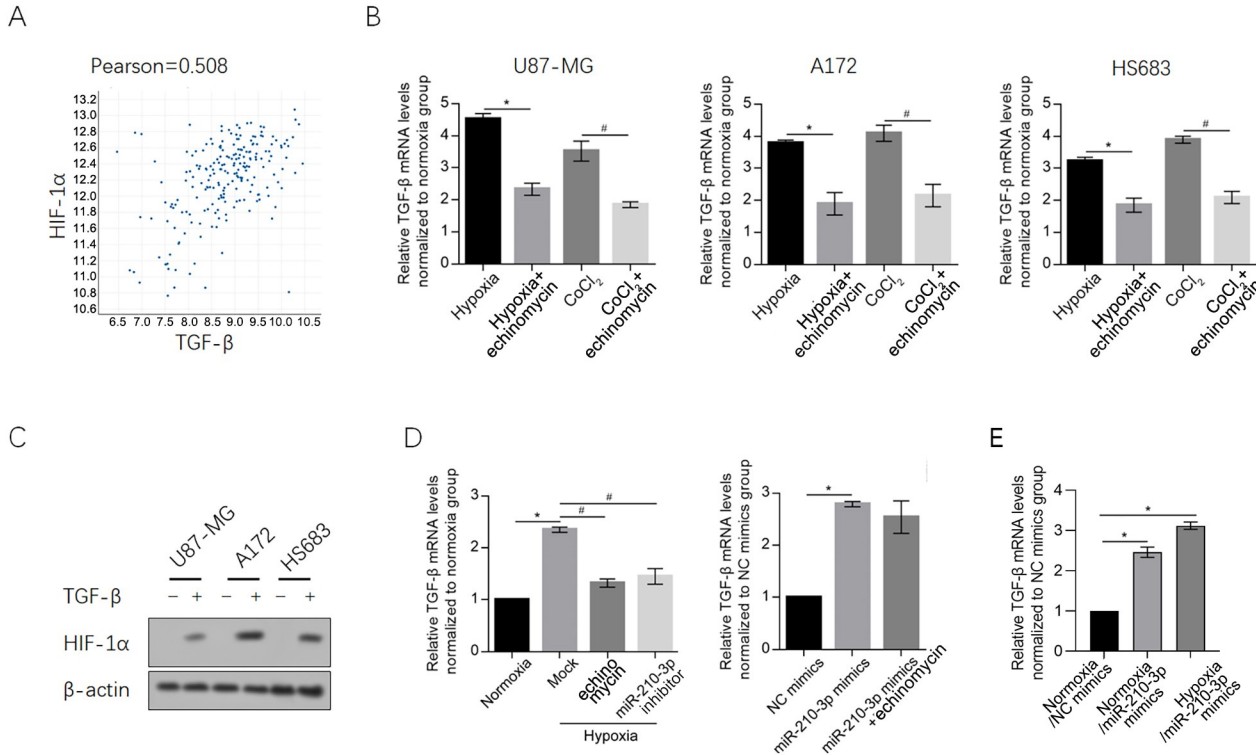

**Fig 3. HIF-1α transcriptionally upregulated TGF-β in glioma cells.** A. The correlation of HIF-1α expression with TGF-β. Data were analyzed using the REMBRANDT database. Increased HIF-1α expression levels were associated with increased expression levels of TGF-β. B. The expressing levels of TGF-β is detected after hypoxia was detected by RT-qPCR. *P<0.05, vs. hypoxia group; #P<0.05, vs. CoCl₂ group. C. HIF-1α was semi-quantitatively measured by western blot after TGF-β exposure. *P<0.05, vs. normoxia group; #P<0.05, vs. hypoxia group. D. After introduction of miR-210-3p mimics or inhibitor, the mRNA levels of TGF-β are detected by RT-qPCR. *P<0.05, vs. NC mimics group. E. After hypoxia exposure with or without miR-210-3p mimics introduction, the mRNA levels of TGF-β are detected by RT-qPCR. *P<0.05, vs. NC mimics group.

## Hypoxia-induced miR-210-3p promoted EMT potentially via upregulating TGF-β1

We then aim to figure out whether miR-210-3p induced by hypoxia exposure affects EMT, which is promoted by TGF-β expression [29]. hallmarkers of EMT, including E-cadherin, N-cadherin and Vimentin were detected by western blot. As it is shown in Fig 5A (Left panel), Hypoxia exposure obviously promoted EMT, which was reversed by echinomycin co-treatment, indicated that transcriptional activity of HIF-1α induced by hypoxia is critical for EMT promotion. Meanwhile, the similar effect on EMT was also observed in miR-210-3p inhibitor introducing group, demonstrated that miR-210-3p transcriptionally induced by HIF-1α potentially plays critical role in promoting EMT after hypoxia exposure. To evaluate the effect of miR-210-3p on EMT, hallmarkers of EMT were detected under normoxia condition after miR-210-3p mimics introduction. As it is presented in Fig 5A (right panel), introduction of miR-210-3p mimics obviously promoted EMT, which is similar to the effect of addition of 5 ng/ml of TGF-β. Hypoxia exposure converted a cobblestone-like or a short spindle-shaped epithelial profile to a stick-like or long spindle shaped mesenchymal morphology in glioma cells, which was reversed by addition of echinomycin or inhibitor of miR-210 (Fig 5B). These results suggested that miR-210-3p plays critical roles in EMT promotion, which is potentially via regulating TGF-β expression. Subsequently, invasive ability of cells was evaluated by performing Transwell assay, and consistent with the profiling changes of hallmarkers of EMT, invasive

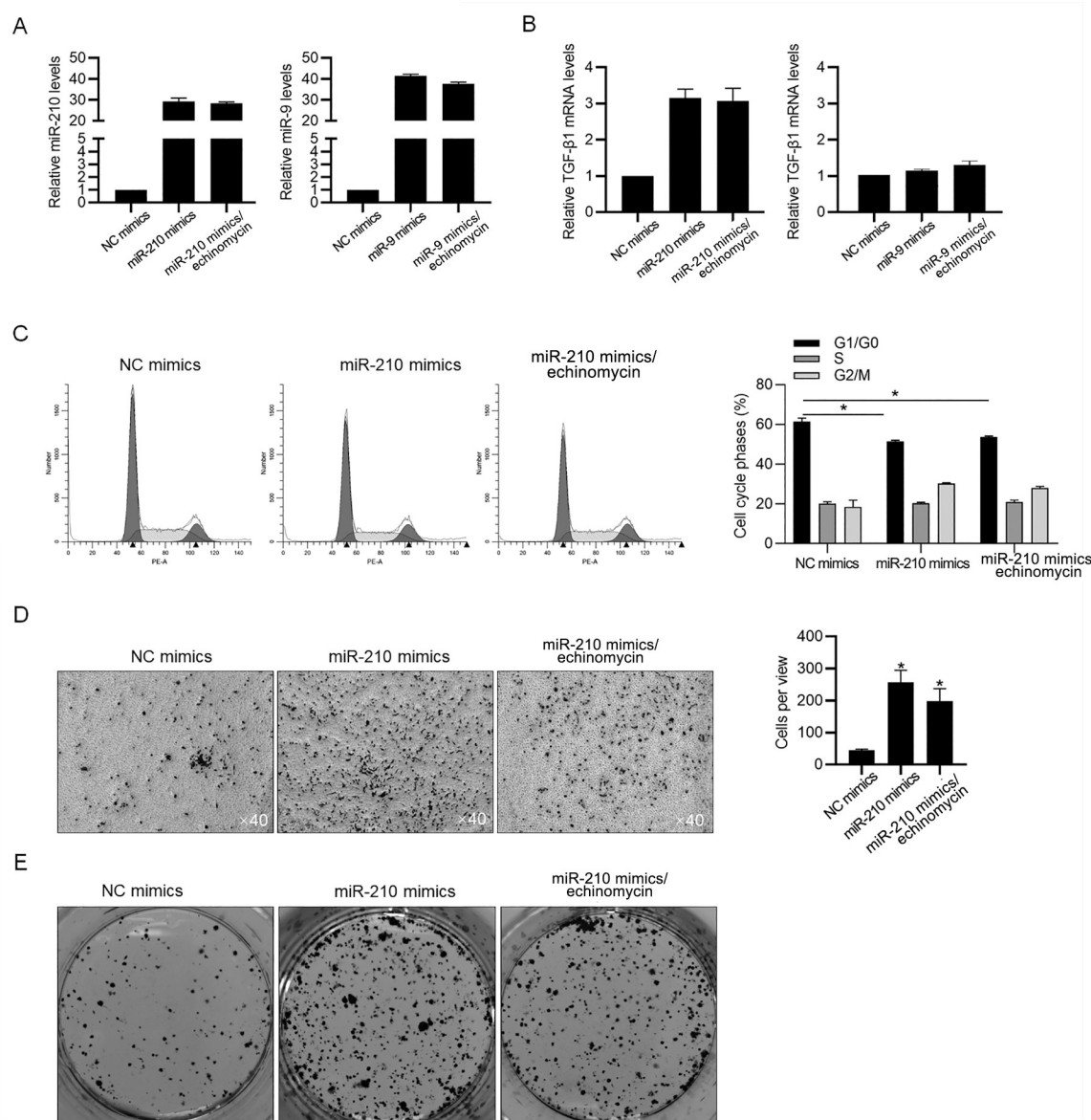

**Fig 4. miR-210 mimics upregulated TGF-β1 transcriptionally and promoted proliferation, invasion and colony formation abilities.**
A. After miR-210 mimics, or miR-9 mimics were transfected, RT-qPCR was performed to detect the amount of overexpressed miR-210 or miR-9. B. The TGF-β1 mRNA level was measured by performing RT-qPCR. C. Cell cycle was measured by performing PI staining followed by flow cytometric assay. *P<0.05, NC mimics group. D. Invasion was measured by performing transwell assay. *P<0.05, NC mimics group. E. Colony formation was performed to detect the tumor formation in vitro.

ability presented same tendency expectedly, which further confirmed the effects of miR-210-3p on EMT (Fig 5C).

## MiR-210-3p induced chemoresistance in U87-MG potentially via upregulating TGF-β

It is reported that hypoxia-induced TGF-β is positively associated with chemoresistance in cancer cells [26, 27], this promoted us to evaluate the effect of miR-210 and subsequent stimulated TGF-β on chemoresistance in U87-MG cells. After introduction of miR-210-3p mimics,

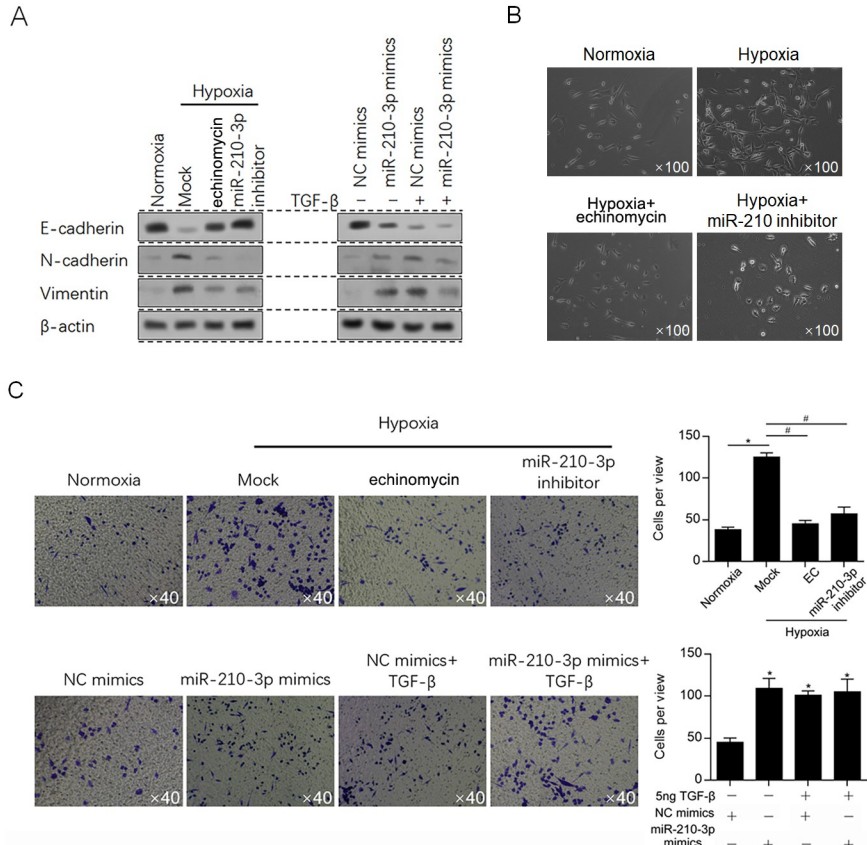

**Fig 5. miR-210-3p promoted EMT potentially via regulating TGF-β.** A. Western blot was performed to detect hallmarkers of EMT, including E-cadherin, N-cadherin and Vimentin. B. Morphological presentation of glioma U87MG cells after hypoxia exposure. C. Transwell assay was performed to detect the effects of miR-210-3p on invasive ability. For upper panel, *P<0.05, vs. Normoxia group; #P<0.05, vs. Hypoxia group. For lower panel, *P<0.05, vs. NC mimics group.

cells were treated with a range concentration of TMZ from 7.5 to 480 μM for 24 h and analyzed by performing CCK-8 assay. As it is shown in Fig 6A (left panel), miR-210-3p introduction obviously desensitized U87-MG to TMZ, which is similar to that of treatment of 5 ng/ml of TGF-β (right panel). This demonstrated that both miR-210-3p introduction and TGF-β treatment promoted chemoresistance in U87-MG. To further confirm whether miR-210-3p promotes chemoresistance by upregulating TGF-β, TGF-β was knockdown in miR-210-3p-introduced cells, and the results presented that TGF-β knockdown obviously sensitized U87-MG to TMZ (Fig 6B). Then, the apoptotic death rate after 100 μM TMZ treatment was performed, and expectedly, miR-210-3p significantly decreased apoptotic cell death, which was reversed by TGF-β knockdown (Fig 6C).

## Hypoxia-induced miR-210-3p partially regulates U87-MG via activating NF-κB signalling pathway

To investigate the underlying mechanism of the regulatory role of miR-210-3p induced by hypoxia exposure in U87MG, we found that hypoxia exposure significantly increased, while addition of echinomycin or QNZ reduced NF-κB-dependent luciferase activity in U87MG cells, which indicated that hypoxia-activated NF-κB is, at least partially, depending on HIF-1α

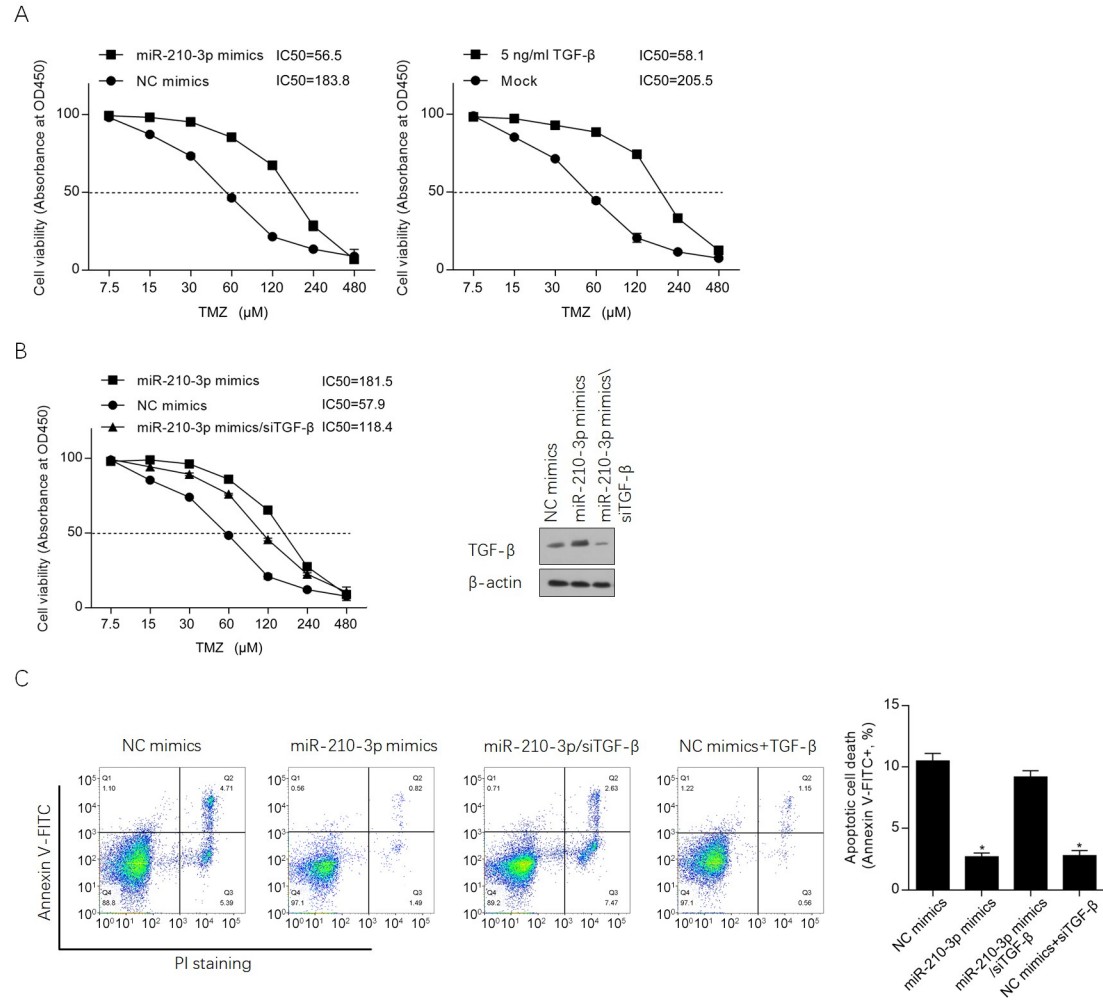

**Fig 6. miR-210-3p induced chemoresistance potentially via regulating TGF-β.** A. After introduction of miR-210-3p mimics, or treated with 5 ng/ml of TGF-β for 24 h, chemosensitivity to TMZ was evaluated by performing CCK-8 assay. B. After knockdown of TGF-β, chemosensitivity to TMZ was evaluated by performing CCK-8 assay. C. Apoptotic cell death was measured by performing Annexin V-FITC/PI double staining followed by flow cytometry after 24-hour treated with 100 μM TMZ. *P<0.05, vs. NC mimics group.

activity (Fig 7A). To confirm whether miR-210-3p was involved in activation of NF-κB pathway, we introduced miR-210 mimics with or without addition of QNZ. Expectedly, miR-210 mimics significantly activated NF-κB pathway, which was reversed by addition of QNZ. Moreover, cellular fractionation and western blot analysis revealed that hypoxia exposure enhanced, while addition of echinomycin or QNZ reduced nuclear accumulation of NF-κB/p65 (Fig 7C). By performing RT-qPCR analysis, it was further showed that hypoxia exposure increased the expression levels of multiple NF-κB signalling downstream metastasis-related target genes, including TWIST1, MMP13 and IL11 in U87MG cells (Fig 7D). We then analysed invasion ability and found that hypoxia-activated NF-κB promoted cell invasion in U87MG (Fig 7E).

## Discussion

Our study revealed that hypoxia-dependent miR-210-3p induction is transcriptionally upregulated by HIF-1α and that it positively increased TGF-β expression of in glioma cells.

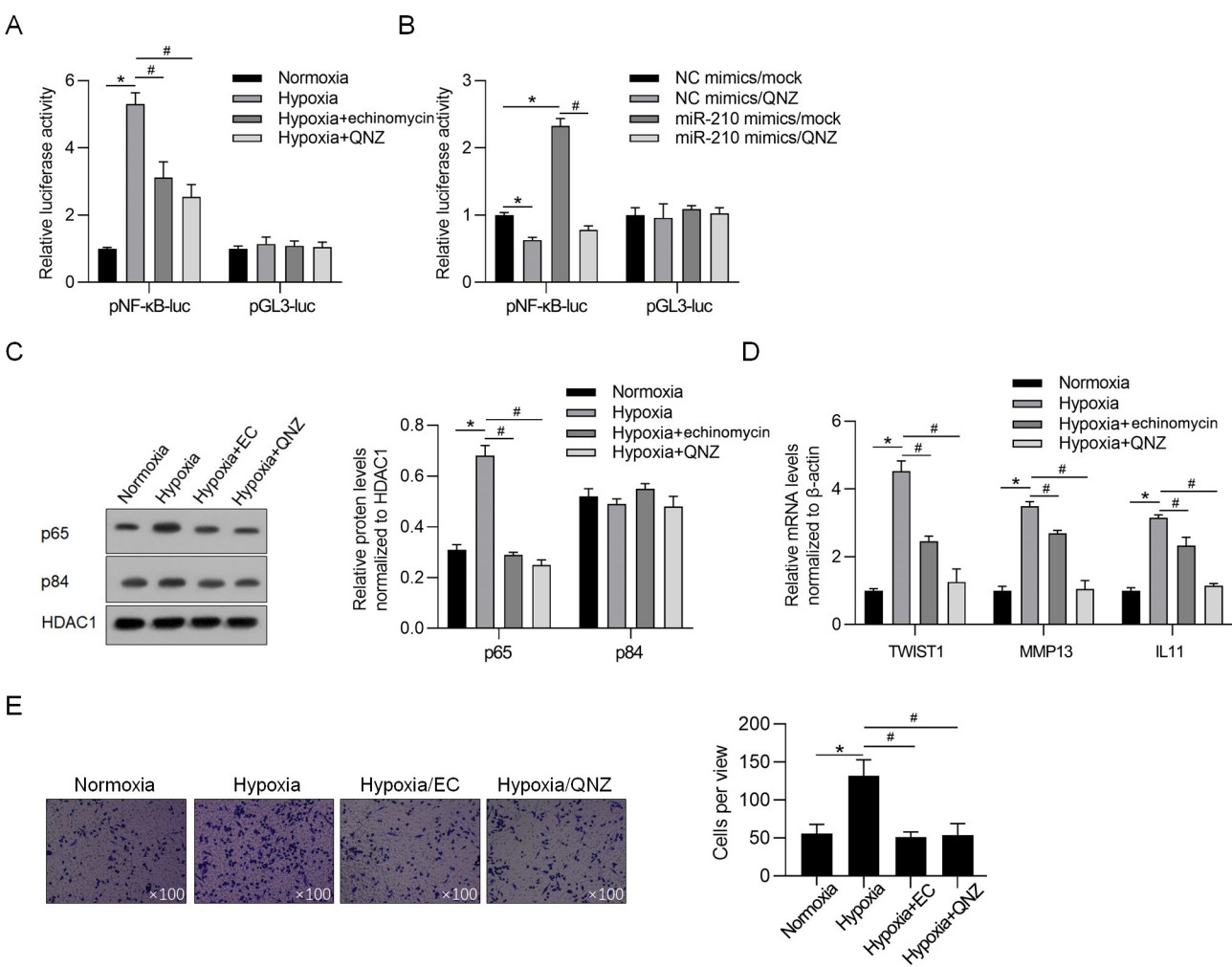

**Fig 7. Hypoxia-induced miR-210 activates NF-κB signaling pathway and promotes invasion.** A. After hypoxia exposure with echinomycin or QNZ, the transcriptional activity of NF-κB was measured by performing luciferase reporter assay. *P<0.05, vs. Normoxia group; #P<0.05, vs. Hypoxia group. B. After introduction of miR-210 mimics, with or without QNZ, the transcriptional activity of NF-κB was measured by performing luciferase reporter assay. *P<0.05, vs. NC mimics/mock group; #P<0.05, vs. miR-210 mimics/mock group. C. Western blot was performed to detect nuclear NF-κB/p65 expression. The nuclear p84 and HDAC1 were used as the nuclear protein markers. D. RT-qPCR analysis of TWIST1, MMP13 and IL11 was performed after hypoxia exposure with addition of echinomycin or QNZ. *P<0.05, vs. Normoxia group; #P<0.05, vs. Hypoxia group. E. Invasion ability was performed to detect the effect of hypoxia on invasion via activating NF-κB signaling. *P<0.05, vs. Normoxia group; #P<0.05, vs. Hypoxia group.

Subsequently, this TGF-β upregulation potentially promotes EMT, invasive ability and chemoresistance to TMZ. Interestingly, TGF-β is responsible for upregulating HIF-1α, which indicates a feedback loop between HIF-1α and TGF-β. These results indicate that hypoxia-induced miR-210-3p expression may act as an oncogene to promote invasiveness by promoting EMT to protect in glioma cells from chemotherapy.

Accumulating evidence has revealed that a group of hypoxia-relevant microRNAs are tightly associated with the oxidative stress response and subsequent tumor progression in several kinds of cancer cells [30]. Li and colleagues reported that miR-137, a novel hypoxia-responsive microRNA, is tightly involved in the regulation of tumor progression and that it exerts protective effects in vivo by inhibiting mitophagy by targeting a key regulator of mitophagy [31]. It has also been reported that miR-101 is responsible for hypoxia exposure and that it promotes angiogenesis via the heme oxygenase-1/vascular endothelial growth factor axis by targeting cullin3, indicating that it has protective roles in human umbilical vein

endothelial cells [32]. All of these data inspired us to dissect the exact roles of microRNAs in glioma cells.

Based on our results, it appears that HIF-1α induction via both hypoxia exposure and CoCl$_2$ treatment transcriptionally upregulated miR-210-3p, revealing the specific regulatory mechanism of miR-210-3p under hypoxic conditions. To address our aim of revealing the effects of miR-210-3p on hypoxia-induced EMT and invasiveness in glioma cells, we treated hypoxia-exposed cells with echinomycin or introduced a miR-210-3p inhibitor (Fig 3B). We observed that both of these two treatments had similar effects on invasiveness, indicating that transcriptional induction of miR-210-3p may play critical roles in these processes. We also evaluated the effects of miR-210-3p on other malignant behaviors, including proliferation, colony formation and tumor formation; however, while echinomycin treatment clearly affected these phenomena, the miR-210-3p inhibitor had no effect, indicating that miR-210-3p is mainly involved in the regulation of invasiveness. Echinomycin, a potent small-molecule and cell-permeable inhibitor of hypoxia-inducible factor-1 (HIF-1) DNA-binding activity [33], was employed to inhibit HIF-1 transcriptional activity and thus to investigate the regulation of HIF-1α on miR-210-3p. HIF-1α functions as a transcriptional regulator and also interacts with proteins, or even long non-coding RNA and thus exerts multiple functions [34, 35]. By considering this, echinomycin was employed instead of siRNA targeting to HIF-1 mRNA to specifically inhibit HIF-1α transcriptional activity without disturbing other functions.

We also observed that miR-210-3p is positively associated with the TGF-β mRNA level (Fig 2D); however, without knowing the status of TGF-β release into supernatant, the confirmation of our hypothesis was limited. To determine whether TGF-β is involved in miR-210-3p-induced malignancy, glioma cells were cultured with TGF-β supplementation under normoxic conditions, and we observed that miR-210-3p-promoted invasiveness was similar to the effects of TGF-β addition. By measuring apoptotic cell death under hypoxic conditions, we found that miR-210-3p protected the cells from TMZ-induced apoptosis, which is also similar to the effect of TGF-β supplementation, and this effect was reversed by TGF-β knockdown with a TGF-β-targeted siRNA. These data suggest that miR-210-3p exerts its malignancy-promoting effects in glioma cells via a mechanism mainly dependent on TGF-β expression, which is consistent with the previous report showing that miR-210-3p exerts as a promoter of malignancies in glioma [36]. However, we failed to further confirm the effects of miR-210-3p induced TGF-β on cellular behaviors by adding TGF-β neutralizing antibody, which is a limitation in this study and is worth doing in further study.

Previous work showed that hypoxia-responsive miR-210-3p modulates mitochondrial respiration in the placenta by targeting mitochondrial-related genes [37]. In particular, miR-210-3p modified the maintenance of mitochondrial membrane potential to improve mitochondrial function, which is a primary promoting factor of cancer progression. According to our results, despite of the roles of miR-210-3p on mitochondrial functions, HIF-1α induced miR-210-3p may also promote malignant behaviors, including invasiveness and EMT in glioma cells, which may also be involved in the regulation of mitochondrial function.

## Supporting information

**S1 File.**
(ZIP)

## Acknowledgments

The authors would like to thank Mr. Tao Hong for language editing and suggestions for statistical analysis.

## Author Contributions

**Conceptualization:** Qiongying Hu.

**Data curation:** Changjin Chen.

**Formal analysis:** Qiongying Hu.

**Funding acquisition:** Ziyi Zhao.

**Investigation:** Hong Liu, Changjin Chen, Jinhao Zeng.

**Methodology:** Hong Liu, Changjin Chen, Ziyi Zhao, Qiongying Hu.

**Resources:** Jinhao Zeng.

**Software:** Hong Liu, Jinhao Zeng.

**Supervision:** Ziyi Zhao.

**Writing – original draft:** Ziyi Zhao, Qiongying Hu.

**Writing – review & editing:** Ziyi Zhao, Qiongying Hu.

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
