## [Editor Report · Decision Letter 0]

25 Jan 2021

PONE-D-21-00617

MicroRNA-210-3p is transcriptionally upregulated by hypoxia induction and thus promoting EMT and chemoresistance in glioma cells

PLOS ONE

Dear Dr. Zhao,

Thank you for submitting your manuscript to PLOS ONE. After careful consideration, we feel that it has merit but does not fully meet PLOS ONE’s publication criteria as it currently stands. Therefore, we invite you to submit a revised version of the manuscript that addresses the points raised during the review process.

Before I send the manuscript out for review, there are number of points that I think need to be clarified. 

In the methods, what exactly does cells "were stored immediately to avoid being passaged for too many times" mean - how many times were they passaged?. The levels of hypoxia under various culture conditions need to be specified.  Are the cultures actually anoxic?  How are final O2 levels measured?  How do they compare with O2 levels in situ, that is, within a tissue? This speaks to the physiological relevance of these studies. The use of CoCl2 treatment to mimic hypoxia seems problematic, not withstanding its convenience.  Do the authors have comparative RNA SEQ analyses (for example) to establish how closely CoC2 treatment and hypoxia conditions resemble each other?  This seems necessary given that "CoCl2 influences the transcription of distinct sets of genes that were not affected by low oxygen‐induced hypoxia, indicating that genes induced by both models of hypoxia do not overlap (Vengellur et al., 2005)." (from Muñoz‐Sánchez and Chánez‐Cárdenas, 2019). The authors should use the full name echinomycin rather than the non-standard abbreviation (EC)  and should note that "echinomycin strongly inhibits the activity of HIF‐1 under hypoxic conditions, and also interferes with the activity of other transcription factors. These results demonstrate the lack of specificity of this molecule. Moreover, it is demonstrated that echinomycin induces an increase in HIF‐1 activity under normoxic conditions, parallel to an increase in the expression of HIF‐1 target genes. (Vlaminck et al., 2007)The authors should present their studies in the context of related work on miRNAs and C6 glioma cells (He et  al., 2020. "MiR‑210‑3p Inhibits Proliferation and Migration of C6 Cells by Targeting Iscu". Neurochemical research.)and probably should have cited: Malzkorn et al. 2010) Identification and functional characterization of microRNAs involved in the malignant progression of gliomas. Brain Pathol 20:539–550

We look forward to receiving your revised manuscript.

Kind regards,

Michael Klymkowsky, Ph.D.

Academic Editor

PLOS ONE
---

## [Author Response · Author response to Decision Letter 0]

16 Mar 2021

1. Thank you for providing the following data availability statement:

"All relevant data are within the manuscript and its Supporting Information files."

However, we note you have mentioned the following in your manuscript:

The datasets used during the present study are available from the corresponding author upon

reasonable request."

* Can you please clarify which statement is correct?

Answer：dear Sir, the statement “All relevant data are within the manuscript and its Supporting Information files.” Is correct. We modified the statement in manuscript as “All data are fully available without restriction” in “Availability of data and materials” section

---

## [Decision Letter · Decision Letter 1]

24 May 2021

PONE-D-21-00617R1

MicroRNA-210-3p is transcriptionally upregulated by hypoxia induction and thus promoting EMT and chemoresistance in glioma cells

PLOS ONE

Dear Dr. Zhao,

Thank you for submitting your manuscript to PLOS ONE. After careful consideration, we feel that it has merit but does not fully meet PLOS ONE’s publication criteria as it currently stands. Therefore, we invite you to submit a revised version of the manuscript that addresses the points raised during the review process.

Please address review #1 comments in detail, you can indicate areas of future study and caution, but I do not think new experiments are necessary.  

We look forward to receiving your revised manuscript.

Kind regards,

Michael Klymkowsky, Ph.D.

Academic Editor

PLOS ONE

Journal Requirements:

Reviewers' comments:

Reviewer's Responses to Questions

**Comments to the Author**

1. If the authors have adequately addressed your comments raised in a previous round of review and you feel that this manuscript is now acceptable for publication, you may indicate that here to bypass the “Comments to the Author” section, enter your conflict of interest statement in the “Confidential to Editor” section, and submit your "Accept" recommendation.

Reviewer #1: (No Response)

Reviewer #2: All comments have been addressed

2. Is the manuscript technically sound, and do the data support the conclusions?

Reviewer #1: Partly

Reviewer #2: Yes

3. Has the statistical analysis been performed appropriately and rigorously? 

Reviewer #1: Yes

Reviewer #2: Yes

4. Have the authors made all data underlying the findings in their manuscript fully available?

Reviewer #1: Yes

Reviewer #2: Yes

5. Is the manuscript presented in an intelligible fashion and written in standard English?

Reviewer #1: Yes

Reviewer #2: Yes

6. Review Comments to the Author

Reviewer #1: The manuscript by Liu et al. is an interesting addition to the field. The authors report that hypoxia results in the up-regulation of miRNA-210-3p through the induction of HIF-1alpha and that miRNA-210-3p expression influences EMT and chemoresistance in glioma cells through a mechanism that involves the regulation of TGF-beta. While overall this is a good study, some points should be further addressed.

Major:

1. The authors do a nice job of demonstrating that miR-210-3p is up-regulated with hypoxia. While the data using EC would suggest this up-regulation is due to HIF-1a, some additional evidence, beyond the use of EC, is needed. What happens to miR-210 expression under hypoxic conditions when HIF-1a is knocked-down. Does the miR-210-3p promoter have a HIF-1a binding site(s)?

2. As regards the regulation of TGF-beta, the authors clearly show an involvment of both TGF-beta and miR-210 in the hypoxic response, but no evidence is presented to directly link these molecules. How do the authors propose that miR-210 is regulating TGF-beta? If miR-210-3p is regulating TGF-beta, then addition of TGF-beta to the media of cells in which TGF-beta mRNA and miR-210-3p have been knock-down, should have the same or similar cellular effects as HIF-1alpha stimulation.

3. Does inhibition of TGF-beta and NF-kB block the observed cellular effects of HIF-1alpha? The authors state in the discussion, "that miR-210-3p is positively associated with the TGF-beta mRNA level; however, without knowing the status of TGF-beta release into supernatant, the confirmation of our hypothesis was limited". The authors should assay both the synthesis/stability of TGF-beta mRNA as well as the levels of TGF-beta in the media following hypoxia, in the presence or absence of miR-210-3p. Also use of neutralizing TGF-beta antibody could help determine how much of the effects contributed by miR-210-3p are mediated through TGF-beta.

4. Use of a second relevant chemotherapeutic agent is recommended to enhance the significance of the findings described in U87MG cells.

5. The X-axis of graphs in the same panel (Figures 1 and 4) should be on the same scale.

6. In Figue 3D, it would be helpful to show the effects of adding miR-210-3p to hypoxic cells on TGF-beta mRNA.

7. In Figure 4C, the addition of the miR-210 mimics results in a reduction of cells in G1/Go but no significant increase in S or G2 is visible. Is there an increase in the sub-G1/G0 population? Additionally, treatment with EC causes an increase in G2, but in the absence of an EC only control it is difficult to determine if miR-210-3p has any influence.

8. In Figure 5C, how do the authors explain the lack of an additive effect of adding miR210-3p mimics and TGF-beta.

Minor:

1. There are several grammatical errors that need to be addressed.

2. As presented by the Editorial Manager, many of the figures were out of order and not of good quality making them difficult to follow.

Reviewer #2: (No Response)

7. PLOS authors have the option to publish the peer review history of their article (what does this mean?). If published, this will include your full peer review and any attached files.

Reviewer #1: No

Reviewer #2: No

---

## [Author Response · Author response to Decision Letter 1]

28 May 2021

Reviewer #1: The manuscript by Liu et al. is an interesting addition to the field. The authors report that hypoxia results in the up-regulation of miRNA-210-3p through the induction of HIF-1alpha and that miRNA-210-3p expression influences EMT and chemoresistance in glioma cells through a mechanism that involves the regulation of TGF-beta. While overall this is a good study, some points should be further addressed.

Major:

1. The authors do a nice job of demonstrating that miR-210-3p is up-regulated with hypoxia. While the data using EC would suggest this up-regulation is due to HIF-1a, some additional evidence, beyond the use of EC, is needed. What happens to miR-210 expression under hypoxic conditions when HIF-1a is knocked-down. Does the miR-210-3p promoter have a HIF-1a binding site(s)?

Answer: Thanks for the suggestion. HIF-1a functions as a master transcriptional regulator of the adaptive response to hypoxia. Under hypoxic conditions, activates the transcription of numerous genes. Meanwhile, HIF-1a protein interacts with redox regulatory protein APEX1 and thus activates NCOA1 and CREBBP. In the purpose of investigating the transcriptional role of HIF-1a on miR-210-3p, we employed EC co-treatment, instead of HIF-1a knockdown which will disturb the interaction between HIF-1a and APEX1.

It is still don’t known whether HIF-1a directly binds to miR-210-3p promoter region. In further investigation, it is worth performing ChIP-seq to screen HIF-1a binding site(s) in promoter region of HIF-1a.

2. As regards the regulation of TGF-beta, the authors clearly show an involvment of both TGF-beta and miR-210 in the hypoxic response, but no evidence is presented to directly link these molecules. How do the authors propose that miR-210 is regulating TGF-beta? If miR-210-3p is regulating TGF-beta, then addition of TGF-beta to the media of cells in which TGF-beta mRNA and miR-210-3p have been knock-down, should have the same or similar cellular effects as HIF-1alpha stimulation.

Answer: No evidence was shown to present the direct link between TGF-b and miR-210. In figure 3B, inhibition of HIF-1a activity by EC addition decreased TGF-b mRNA. This indicated that HIF-1a activity upregulate TGF-B mRNA. In figure 4B, it is also showed that introduction of miR-210 mimics upregulated TGF-b mRNA, which is not reversed by EC co-treatment. This indicates that HIF-1a upregulated TGF-b mRNA dependent on miR-210-3p. In figure 5c, addition of TGF-b present similar effect with miR-210 mimics on promoting invasion. These results indirectly indicated that HIF-1a regulates TGF-b via regulating miR-210-3p. 

3. Does inhibition of TGF-beta and NF-kB block the observed cellular effects of HIF-1alpha? The authors state in the discussion, "that miR-210-3p is positively associated with the TGF-beta mRNA level; however, without knowing the status of TGF-beta release into supernatant, the confirmation of our hypothesis was limited". The authors should assay both the synthesis/stability of TGF-beta mRNA as well as the levels of TGF-beta in the media following hypoxia, in the presence or absence of miR-210-3p. Also use of neutralizing TGF-beta antibody could help determine how much of the effects contributed by miR-210-3p are mediated through TGF-beta.

Answer: In figure 6, figure 7, inhibition of TGF-beta and NF-kB block the observed cellular effects of HIF-1alpha. Thanks for the precious suggestion. In this study, we mainly focused on the regulatory effects of miR-210-3p on TGF-b mRNA, so more attention was put into the identification of regulatory effect of miR-210-3p on TGF-b mRNA. In further research, we’d like to quantitatively measure the contribution of miR-210-3p on cellular behavior through TGF-b.

We’ve also discussed this as a limitation in this study.

4. Use of a second relevant chemotherapeutic agent is recommended to enhance the significance of the findings described in U87MG cells.

Answer: Temozolomide (TMZ) is an alkylating agent currently used as first-line therapy in standard treatment of GBM. Temozolomide (TMZ) chemotherapy has been widely accepted as the new standard of care for patients with newly diagnosed GBM. That’s why we employed TMZ in this study.

5. The X-axis of graphs in the same panel (Figures 1 and 4) should be on the same scale.

Answer: It has been modified.

6. In Figue 3D, it would be helpful to show the effects of adding miR-210-3p to hypoxic cells on TGF-beta mRNA.

Answer: It has been added.

7. In Figure 4C, the addition of the miR-210 mimics results in a reduction of cells in G1/Go but no significant increase in S or G2 is visible. Is there an increase in the sub-G1/G0 population? Additionally, treatment with EC causes an increase in G2, but in the absence of an EC only control it is difficult to determine if miR-210-3p has any influence.

Answer: Sorry for the mistake in figure 4C (right panel). The sub G2/M was incorrectly calculated via flow cytometry results. It is re-calculated! 

In figure 4c, miR-210/echinomycin group was compared with miR-210 group, the populations of G1/G0, S, or G2/M were not obviously different, this indicated that in cells transfected with miR-210 mimics was not affected by echinomycin addition.

8. In Figure 5C, how do the authors explain the lack of an additive effect of adding miR210-3p mimics and TGF-beta.

Answer: The addition of miR-210-3p mimics is sufficient to stimulate invasion. Added TGF-β is no longer enhance this cellular behavior. 

Minor:

1. There are several grammatical errors that need to be addressed.

Answer: The grammatical errors have been checked carefully.

2. As presented by the Editorial Manager, many of the figures were out of order and not of good quality making them difficult to follow.

Answer: The figures were modified for a better understanding. Thanks for the suggestion.

---

## [Decision Letter · Decision Letter 2]

3 Jun 2021

PONE-D-21-00617R2

MicroRNA-210-3p is transcriptionally upregulated by hypoxia induction and thus promoting EMT and chemoresistance in glioma cells

PLOS ONE

Dear Dr. Zhao,

Thank you for submitting your manuscript to PLOS ONE. After careful consideration, we feel that it has merit but does not fully meet PLOS ONE’s publication criteria as it currently stands. Therefore, we invite you to submit a revised version of the manuscript that addresses the points raised during the review process.

Please respond to the reviewer's specific comments and outline those responses for me..

We look forward to receiving your revised manuscript.

Kind regards,

Michael Klymkowsky, Ph.D.

Academic Editor

PLOS ONE

Journal Requirements:

Reviewers' comments:

Reviewer's Responses to Questions

**Comments to the Author**

1. If the authors have adequately addressed your comments raised in a previous round of review and you feel that this manuscript is now acceptable for publication, you may indicate that here to bypass the “Comments to the Author” section, enter your conflict of interest statement in the “Confidential to Editor” section, and submit your "Accept" recommendation.

Reviewer #1: All comments have been addressed

2. Is the manuscript technically sound, and do the data support the conclusions?

Reviewer #1: Yes

3. Has the statistical analysis been performed appropriately and rigorously? 

Reviewer #1: Yes

4. Have the authors made all data underlying the findings in their manuscript fully available?

Reviewer #1: Yes

5. Is the manuscript presented in an intelligible fashion and written in standard English?

Reviewer #1: Yes

6. Review Comments to the Author

Reviewer #1: Most comments have been adequately addressed. Previous Comment 1 was directed at the specificity of EC. Many "specific inhibitors" are not so specific and if knock-down of the supposed target is not employed the authors should discuss briefly the limitations of EC (i.e., what is its specificity--IC50--for HIF and whether it is known to affect other enzymes; in order for the reader to have a full appreciation of the possible limitations and implications of using this methodology. As regards previous Comment 2, the authors should make sure that the train of thought and interpretation of the findings as presented in their response to the Reviewer's comment is clearly made to the reader in the manuscript.

All other points were sufficiently addressed.

7. PLOS authors have the option to publish the peer review history of their article (what does this mean?). If published, this will include your full peer review and any attached files.

Reviewer #1: No

---

## [Author Response · Author response to Decision Letter 2]

4 Jun 2021

"Please respond to the reviewer's specific comments and outline those responses for me.."

The reviewer comments, in section 6 are:

"Reviewer #1: Most comments have been adequately addressed. Previous Comment 1 was directed at the specificity of EC. Many "specific inhibitors" are not so specific and if knock-down of the supposed target is not employed the authors should discuss briefly the limitations of EC (i.e., what is its specificity--IC50--for HIF and whether it is known to affect other enzymes; in order for the reader to have a full appreciation of the possible limitations and implications of using this methodology. As regards previous Comment 2, the authors should make sure that the train of thought and interpretation of the findings as presented in their response to the Reviewer's comment is clearly made to the reader in the manuscript.

Answer: Echinomycin (Quinomycin A) is potent small-molecule and cell-permeable inhibitor of hypoxia-inducible factor-1 (HIF-1) DNA-binding activity. In discussion section, we discussed the reason why echinomycin was employed instead of siRNA targeting to HIF-1 mRNA as a limitation of this manuscript. The methodology of this choice was explained to make it more clear for reader to have a full appreciation of this limitation. The modified section was described as follows (3rd paragraph, discussion section): 

Based on our results, it appears that HIF-1α induction via both hypoxia exposure and CoCl2 treatment transcriptionally upregulated miR-210-3p, revealing the specific regulatory mechanism of miR-210-3p under hypoxic conditions. To address our aim of revealing the effects of miR-210-3p on hypoxia-induced EMT and invasiveness in glioma cells, we treated hypoxia-exposed cells with echinomycin or introduced a miR-210-3p inhibitor (Figure 3B). We observed that both of these two treatments had similar effects on invasiveness, indicating that transcriptional induction of miR-210-3p may play critical roles in these processes. We also evaluated the effects of miR-210-3p on other malignant behaviors, including proliferation, colony formation and tumor formation; however, while echinomycin treatment clearly affected these phenomena, the miR-210-3p inhibitor had no effect, indicating that miR-210-3p is mainly involved in the regulation of invasiveness. Echinomycin, a potent small-molecule and cell-permeable inhibitor of hypoxia-inducible factor-1 (HIF-1) DNA-binding activity [33], was employed to inhibit HIF-1 transcriptional activity and thus to investigate the regulation of HIF-1α on miR-210-3p. HIF-1α functions as a transcriptional regulator and also interacts with proteins, or even long non-coding RNA and thus exerts multiple functions [34,35]. By considering this, echinomycin was employed instead of siRNA targeting to HIF-1 mRNA to specifically inhibit HIF-1α transcriptional activity without disturbing other functions.

All other points were sufficiently addressed."

---

## [Editor Report · Decision Letter 3]

8 Jun 2021

MicroRNA-210-3p is transcriptionally upregulated by hypoxia induction and thus promoting EMT and chemoresistance in glioma cells

PONE-D-21-00617R3

Dear Dr. Zhao,

We’re pleased to inform you that your manuscript has been judged scientifically suitable for publication and will be formally accepted for publication once it meets all outstanding technical requirements.

Kind regards,

Michael Klymkowsky, Ph.D.

Academic Editor

PLOS ONE
---

## [Editor Report · Acceptance letter]

18 Jun 2021

PONE-D-21-00617R3 

MicroRNA-210-3p is transcriptionally upregulated by hypoxia induction and thus promoting EMT and chemoresistance in glioma cells 

Dear Dr. Zhao:

I'm pleased to inform you that your manuscript has been deemed suitable for publication in PLOS ONE. Congratulations! Your manuscript is now with our production department. 

Kind regards, 

on behalf of

Dr. Michael Klymkowsky 

Academic Editor

PLOS ONE